# Comparative Transcriptomics of Immune Checkpoint Inhibitor Myocarditis Identifies Guanylate Binding Protein 5 and 6 Dysregulation

**DOI:** 10.3390/cancers13102498

**Published:** 2021-05-20

**Authors:** Daniel Finke, Markus B. Heckmann, Janek Salatzki, Johannes Riffel, Esther Herpel, Lucie M. Heinzerling, Benjamin Meder, Mirko Völkers, Oliver J. Müller, Norbert Frey, Hugo A. Katus, Florian Leuschner, Ziya Kaya, Lorenz H. Lehmann

**Affiliations:** 1Department of Cardiology, Heidelberg University Hospital, 69120 Heidelberg, Germany; Daniel.Finke@med.uni-heidelberg.de (D.F.); Markus.Heckmann@med.uni-heidelberg.de (M.B.H.); Janek.Salatzki@med.uni-heidelberg.de (J.S.); Johannes.Riffel@med.uni-heidelberg.de (J.R.); Benjamin.Meder@med.uni-heidelberg.de (B.M.); Mirko.Voelkers@med.uni-heidelberg.de (M.V.); Norbert.Frey@med.uni-heidelberg.de (N.F.); Hugo.Katus@med.uni-heidelberg.de (H.A.K.); Florian.Leuschner@med.uni-heidelberg.de (F.L.); Ziya.Kaya@med.uni-heidelberg.de (Z.K.); 2Cardio-Oncology Unit, Heidelberg University Hospital, 69120 Heidelberg, Germany; 3German Centre for Cardiovascular Research (DZHK), 69120 Partner Site Heidelberg/Mannheim, Germany; 4Department of Pathology, Heidelberg University Hospital, 69120 Heidelberg, Germany; Esther.Herpel@med.uni-heidelberg.de; 5Department of Dermatology, University Hospital Erlangen, 91054 Erlangen, Germany; Lucie.Heinzerling@uk-erlangen.de; 6Deutsches Zentrum Immuntherapie (DZI), University Hospital Erlangen, 91054 Erlangen, Germany; 7Department of Cardiology, University Hospital Kiel, 24105 Kiel, Germany; Oliver.Mueller@uksh.de; 8German Centre for Cardiovascular Research (DZHK), 24105 Partner Site Kiel/Hamburg/Lübeck, Germany; 9Deutsches Krebsforschungszentrum (DKFZ), 69120 Heidelberg, Germany

**Keywords:** PD1, PDL-1, CTLA4, CD8, virus myocarditis, ICI-associated myocarditis, ICIM, comparative transcriptomics

## Abstract

**Simple Summary:**

Immune checkpoint inhibitors are revolutionizing cancer treatment, but lead to the occurrence of immune related adverse events including ICI-associated myocarditis (ICIM). To date, transcriptional alterations of this rare phenomenon with a high mortality rate are not characterized. 19 ICIM patients at the University Hospitals Heidelberg and Kiel showed diverse clinical presentations. Comparative transcriptomics was able to distinguish ICIM patients from patients with dilated cardiomyopathy or virus-induced myocarditis in the upregulation of 3784 genes. The RNA-based analyses and immunohistology revealed a potential role of an inflammasome-regulating protein, GBP5, as a potential pathomechanism in cardiomyocytes. These alterations may help to diagnose ICIM and potentially enable to identify patients at risk in an early stage.

**Abstract:**

Immune checkpoint inhibitors (ICIs) are revolutionizing cancer treatment. Nevertheless, their increasing use leads to an increase of immune-related adverse events (irAEs). Among them, ICI-associated myocarditis (ICIM) is a rare irAE with a high mortality rate. We aimed to characterize the transcriptional changes of ICIM myocardial biopsies and their possible implications. Patients suspected for ICIM were assessed in the cardio-oncology units of University Hospitals Heidelberg and Kiel. Via RNA sequencing of myocardial biopsies, we compared transcriptional changes of ICIM (*n* = 9) with samples from dilated cardiomyopathy (DCM, *n* = 11), virus-induced myocarditis (VIM, *n* = 5), and with samples of patients receiving ICIs without any evidence of myocarditis (*n* = 4). Patients with ICIM (*n* = 19) showed an inconsistent clinical presentation, e.g., asymptomatic elevation of cardiac biomarkers (hs-cTnT, NT-proBNP, CK), a drop in left ventricular ejection fraction, or late gadolinium enhancement in cMRI. We found 3784 upregulated genes in ICIM (FDR < 0.05). In the overrepresented pathway ‘response to interferon-gamma’, we found guanylate binding protein 5 and 6 (compared with VIM: GBP5 (log2 fc 3.21), GBP6 (log2 fc 5.37)) to be significantly increased in ICIM on RNA- and protein-level. We conclude that interferon-gamma and inflammasome-regulating proteins, such as GBP5, may be of unrecognized significance in the pathophysiology of ICIM.

## 1. Introduction

Immune checkpoint inhibitors (ICIs) are effective cancer therapeutics [1]. These monoclonal antibodies, which are directed against members of the T cell immune pathway (PD-1, PD-L1 or CTLA-4), have shown promising therapeutical outcomes, with approval for more than 17 cancer entities in 2019 and more than 2250 active clinical trials in 2018 [2,3].

However, apart from their great success, their multifaceted immune-related adverse events (irAEs) are challenging [4]. In contrast to most of the other autoimmune side effects (e.g., cutaneous or endocrinological events), myocarditis is a rare side effect with a high mortality rate that necessitates prompt diagnosis and thorough medical treatment [5]. Signs and symptoms of ICI-associated myocarditis (ICIM) are reported to be heterogenous and vary from asymptomatic courses to striking signs of heart failure (HF) or acute coronary syndrome (ACS) [6,7,8,9,10,11,12,13].

Animal models have been used to study the effects of the ICI-associated immune reaction on the heart [14]. Genetic deletion of PD-1 in the BALB/c mouse line resulted in dilated cardiomyopathy (DCM) with accumulation of immunoglobulins on the cardiomyocytes’ surface [15]. In these mice, the authors were not able to find cardiac leukocyte infiltration but found autoantibodies that were directed against troponin I (TnI). However, mice with the genetic MRL background showed an immense CD4+ and CD8+ T cell infiltration of the myocardium [16]. PD-L1 deficiency revealed a similar phenotype in these mice [17]. Mice deficient in CTLA-4 died after five weeks and showed a generalized infiltration by lymphocytes of multiple organs, including the heart [18]. The phenotype heavily relied on the genetic background of the mice. In a non-autoimmune model, prior cardiac damage by irradiation led to cardiac adverse effects and leucocyte infiltration in response to pharmacological PD-1 inhibition [19]. In another study, a combination of coxsackievirus B3 infection and PD-1/PD-L1 inhibitors induced acute myocarditis [20]. Another two studies in which C57/Bl6 mice were treated with PD-1 inhibitors [21] or CTLA-4 inhibitors [22] describe a deterioration in left ventricular function in response to the blockage of the immune checkpoints.

Based on these preclinical data, we suppose that ICIM is mainly associated with at least two distinct pathological alterations: [1] autoantibodies against cardiac proteins, especially against TnI and [2] a direct infiltration of the myocardium by cytotoxic T cells. In patients receiving a combination of nivolumab and ipilimumab, identical T cell clones invaded tumors and the heart, suggesting that cardiomyocytes and the neoplastic cells share epitopes [5]. Comparing the degree of lymphocyte infiltration revealed two groups of ICIM, which correlated with the outcome in an observational study of ten ICIM cases [23]. In relation to cellular rejection after heart transplantation, the authors found comparable grades of lymphocytic infiltration. A specific test to detect ICIM as well as the specific underlying mechanisms remained elusive [24].


In order to better understand the pathology and involved transcriptional changes of ICIM, we investigated the transcriptome of ICIM patients and compared it with biopsies from virus-induced myocarditis (VIM) and DCM patients, as well as with biopsies from ICI patients without evidence of myocarditis. Transcriptional changes were characterized by dysregulation of members of the interferon-gamma-dependent pathway, guanylate binding protein 5 and 6 (GBP5 and GBP6). We propose an involvement of the inflammasome in the pathophysiology of ICIM, which plays a minor role in other cardiac pathologies, such as DCM or VIM.

## 2. Methods

### 2.1. Patients

All presented cases were appointed to University Hospitals Heidelberg or Kiel, Germany. The patients were initially treated with different checkpoint inhibitors due to their malignant disease and presented at the cardio-oncology units with a suspected cardiac adverse event.

Upon further cardiological assessment, subjects were enrolled in a cross-sectional study. The study protocol was approved by the ethics committees of the Medical Faculty of the University Heidelberg (S-286/2017, 390/2011) and the University Kiel (A174/09), respectively.

### 2.2. Definitions and Follow-Up

As internal surveillance standard, creatine kinase and/or hs-cTnT levels were assessed before any ICI administration in the oncological departments. Elevations of hs-cTnT were indicative of a potential ICIM and patients were thoroughly assessed.

The diagnosis was validated according to the criteria of ICIM as previously described [11,25]. The certainty of the diagnosis (definite, probable) is summarized in Table 1. The gold standard for the diagnosis of ICIM is cardiac biopsy revealing leukocytic infiltration, including CD3+ and CD8+ cells. If biopsy was not performed or if the result was negative, cMRI was performed and biomarkers were assessed to either confirm or to reject the diagnosis of ICIM. The occurrence of further autoimmune side effects, including, e.g., myositis or myasthenia gravis, as well as ECG abnormalities and negative angiography results, supported the diagnosis of ICIM in some cases. Patients in the non-myocarditis group received ICIs and were suspected of being in an early phase of ICIM. The following assessment did not reveal neither a definite nor a probable diagnosis of ICI-associated myocarditis. Thus, we classified them as negative ICIM.

As a follow-up, patients were examined in the cardio-oncology outpatient clinic two to four weeks after being discharged from the hospital.

VIM patients were diagnosed according to the ESC consensus paper [26]. Immune cell infiltration was confirmed by CD68 and CD3 positive cells. DCM patients were assessed by cardiac catherization and echocardiography. Further characteristics of VIM and DCM are shown in Table 2.

### 2.3. Clinical Data Acquisition

Patient-specific data were extracted from the electronic medical reports, including ECG, laboratory results, echocardiographic measurements, and cMRI results. The initial laboratory tests were assessed on the day of admission when ICIM was suspected. Peak values were measured either on the subsequent stay in hospital or on follow-up examinations. Cardiac biopsies were assessed in case there was a suspicion of ICIM, and biopsies were stored in formaldehyde for histological assessments and at −80 °C before RNA isolation. In four patients, the evidence of ICIM could not be confirmed. We considered them as non-ICIM patients.

Standard cMRI was performed supine in a 1.5-T Ingenia (1.5-T) or 3-T Ingenia CX (3-T) whole body scanner (Philips Healthcare, Best, The Netherlands). Myocardial edema was assessed using a T2-weighted black blood spectral presaturation with inversion sequence (SPIR) (T2-BB) as well as a T2-weighted multiecho Gradient-Spin-Echo sequence (GraSE) (T2-mapping) in a basal, midventricular, and apical SAX during breath-hold in end-expiration with the following parameters: TR 1 RR interval, number of echo images = 18, flip angle = 90°.

Late gadolinium enhancement (LGE) images were acquired with 10 min of Gadobutrol (Gadovist, Schering, Berlin, Germany) 0.14 mmol/kg body weight (1.5-T) or 0.1 mmol/kg body weight (3-T) employing a T1-weighted inversion recovery-prepared fast gradient echo sequence with an optimized inversion time in cine long axis 2-, 3-, and 4-chamber views as well as SAX cine images.

### 2.4. Histologies

Representative images of histology are shown as hematoxylin and eosin (HE), CD3, or CD8 staining. In order to stain GBP5 and GBP6, sections of the biopsies were deparaffinized by passing them through a series of xylene and ethanol dilutions. The slides were blocked by TSA blocking reagents (PerkinElmer) and were incubated overnight with GBP5 (Atlas HPA028656), GBP6 (Atlas HPA027744), and α-Actinin (Sigma A2172) antibodies. For nuclei staining, the sections were incubated with DAPI for 15 min.

Images of GBP5- and GBP6-immunostainings for quantification were obtained using Zeiss Axio Observer at 63× magnification at a constant exposure time. Exemplary images were obtained by a Leica SP8 microscope at 630× magnification under constant image saturation settings. Ten images per slide were taken. Quantification of signal intensity and co-localization was calculated by an automated workflow in CellProfiler (Version 4.0.7, Broad Institute, Boston, USA) [27]. Exemplary confocal images are shown for α-Actinin, GBP5/6, and a merged channel.

### 2.5. RNA Isolation and Sequencing

RNA was extracted from myocardial biopsies using Qiagen All-Prep DNA/RNA Kit (Qiagen, Venlo, The Netherlands), and libraries were prepared using Illumina TruSeq Stranded Total RNA Kit (Illumina, San Diego, CA, USA) according to the manufactures’ manuals. Sequencing was performed as paired-end sequencing, with a read length of 150 bps on the Illumina NovaSeq 6000 platform.

### 2.6. Bioinformatical Analysis

Raw data files were aligned to the human genome (hg38) using RNA STAR (Version 2.7.3, Cold Spring Harbor Laboratory, Cold Spring Harbor, NY, USA). Reads (reads per kilobase million (RPKM)) were extracted from the mapped files using the bioconductor package Rsubread (Version 3.1, The Walter and Eliza Hall Institute of Medical Research, Victoria, Australia). Differential gene expression was calculated using DESeq2 algorithms (European Molecular Biology Laboratory, Heidelberg, Germany) with a false discovery rate (FDR) < 0.05 (R package version 2.12). The PCA comparison was performed within the DESeq2 package. Gene enrichment analysis was conducted via DOSE (version 3.8.2) and enrichplot (version 3.1) packages. Gene Ontology and KEGG Pathways were adjusted for multiple testing using the Benjamini–Hochberg method.

De novo transcription factor enrichment analysis was performed by HOMER (version 4.11). As background sequences, we used the promoter regions (transcriptional start site −500 bps) of non-regulated transcripts (FDR > 0.25). For visualizations, we used the following R packages: corrplot (version 0.84), ggplot2 (version 3.2.1), and heatmap3 (version 3.0.2). 

We determined the degree of immune cell infiltration and the abundance of stromal cells by use of the R package Microenvironment Cell Populations-counter (MCP-counter), as previously published [28].

RNAseq data of ICI patients are available in the EBI ArrayExpress Database, accession number E-MTAB-8867. The raw data files of DCM and VIM patients are available in the Gene Expression Omnibus Database (GEO-NCBI, accession number: GSE120567) as previously published [29].

### 2.7. Statistical Analysis

Pie charts were illustrated using GraphPad Prism (Version 8.0, GraphPad Software, San Diego, CA, USA). To compare two groups with continuous data, we used the Mann–Whitney U test. A confidence interval of 95% was considered significant. For dichotomic values, the binomial distribution model was used; *p*-values < 0.05 were considered significant. In column diagrams, the individual values are shown in addition to the mean and standard error of the mean (SEM) as error bars.

CMRI T2 maps were generated using CVI cmr^42^ software (Version 5.6.6, Circle Cardiovascular Imaging Inc., Calgary, AB, Canada). Endocardial and epicardial borders were defined manually in T2-mapping images using an offset of 10% to avoid partial-volume effects in the subendocardial and subepicardial layers. An exponential decay curve was fitted to the intensity decline of each pixel within the images obtained from the multiecho sequence. Noise of all images was assumed to be constant and linearized so that the regression coefficient (*R*^2^) was used to exclude accidental values (goodness-of-fit parameter). If *R*^2^ was below the value of 0.99, the corresponding T2 value was not considered for further analysis. The remaining T2 constants were color-coded and T2 values were calculated according to 16 AHA-segments (segment 17 was not included). A regional edema was defined with a cut-off over 60 ms.

Regions with LGE were verified in at least one other orthogonal plane and in the same plane being obtained as a second image after changing the direction of readout.

## 3. Results

### 3.1. ICIM Patients Have a Variable Clinical Phenotype

In total, 19 patients with suspected cardiac irAEs were examined at the cardio-oncology units of the University Hospitals Heidelberg and Kiel. The details of the patients’ characteristics are summarized in Table 1.

In most cases, the reason for being treated with checkpoint inhibitors was either melanoma or non-small-cell lung cancer (NSCLC). Most patients were treated with PD-1 inhibitors (68.4%; nivolumab and pembrolizumab). Combination therapies of PD-1 and CTLA-4 inhibitors were administered in 21.1% of cases (Figure 1A).

ICIM was suspected in cases of either cardiac biomarker elevation (hs-cTnT and/or NT-proBNP), symptoms of ACS and HF, or a combination of both. Median time to onset of cardiac irAEs was 81 days (95% CI: 35.5; 234) from the beginning of ICI therapy. The vast majority of patients showed ECG abnormalities, which were obtained at admission to the hospital (84.2%). Mainly, we observed new onset of ST depressions or T-wave inversions. In a few patients, Holter ECG revealed relevant blocks, e.g., sinoatrial blocks.

In echocardiography, the LVEF was initially preserved in 47.4% of patients, and 31.6% of patients were diagnosed with at least a moderately reduced LVEF at presentation (LVEF < 45%).

Regarding their symptoms, we observed a variation ranging from asymptomatic patients (6/19) to Takotsubo cardiomyopathy, sinoatrial block, acute cardiac decompensation, severe myocarditis without major clinical manifestations, or steroid refractory myocarditis with associated global myositis and myasthenia-like syndrome. Of the 19 patients, 18 showed initially elevated levels of hs-cTnT (15 pg/mL to 1261 pg/mL) and therefore were admitted to an expanded cardiological assessment including myocardial biopsies (11/19) and/or cMRI (12/19). NT-proBNP was elevated in 84.2% of patients, whereas creatine kinase (CK) (63%) and TnI (31.6%) were less frequently elevated. All patients showed signs of cardiac irAEs. Myocardial biopsies verified the diagnosis of ICIM in 8/19 cases. The remaining patients were diagnosed by means of a combinatorial approach of late gadolinium enhancement and/or edema in cMRI, a drop in left ventricular function (LVEF), elevated cardiac biomarkers, ECG abnormalities, and further autoimmune events in non-cardiac organs and tissues. According to the recently suggested criteria of ICIM, 16 patients were classified as definite and 3 patients were classified as probable (Table 1). An exemplary histological staining is shown in Figure 1B. Figure 1C shows an exemplary T2-weighted image of apical subendocardial edema.

Statistically, we did not find any significant difference in the relevant parameters (e.g., clinical outcome or phenotype) between high vs. low cardiac troponin (above/below the median), symptomatic and asymptomatic patients, patients with preserved and reduced LV-function, or patients having myocardial biopsies with or without detection of infiltrating immune cells. Therefore, we were not able to set up cohorts based on the clinical phenotype.

### 3.2. Transcriptional Analysis Reveals a Distinct Expression Pattern in ICI-Associated Myocarditis (ICIM) Compared with Dilated Cardiomyopathy (DCM) and Virus-Induced Myocarditis (VIM)

Due to the inconsistent clinical picture of ICIM, we decided to characterize its cardiac gene expression in more detail. Myocardial biopsies of 15 patients were collected shortly after being suspected of a cardiac irAE. Transcriptional analysis was performed from nine patients with confirmed diagnosis of ICIM. Biopsies from four patients with a rejected diagnosis were sequenced (Figure 1D).

We compared the expression with data from DCM and VIM and found a strong opposing clustering for ICIM (Appendix A). This clustering was primarily defined by a large number of different transcripts when compared with either DCM or VIM (Appendix A). The same stringent statistical parameters only identified a limited number of differentially regulated genes in DCM vs. VIM (Appendix A). The clinical characteristics of VIM and DCM patients are summarized in Table 2.

Differential gene expression analysis found 3784 unique transcripts that were significantly upregulated in ICIM in at least one of two comparisons, either compared with DCM or with VIM. In DCM we found 1652 and in VIM we found 1181 significantly upregulated genes in both comparisons (Appendix A). However, DCM and VIM shared 1631 upregulated genes (Appendix A).

De novo transcription factor analysis of upregulated genes in ICIM revealed an enrichment of several motifs including ETS (*p* = 10^−16^, FOSL1 (*p* = 10^−14^), YY1 (*p* = 10^−12^), SMAD2/3 (*p* = 10^−12^), and IRF2 (*p* = 10^−12^) (Appendix A).

By overlapping both sets of differentially expressed genes between ICIM and DCM as well as ICIM and VIM, we identified a subset of 1125 genes that were upregulated in both comparisons. Particularly, a set of genes including CCDC175 (log2 fc vs. VIM: 5.43; vs. DCM: 4.72), CXCL9 (log2 fc vs. VIM: 4.37; vs. DCM: 5.59), CXCL11 (log2 fc vs. VIM: 3.24; vs. DCM: 4.4), GBP5 (log2 fc vs. VIM: 3.21; vs. DCM: 4.76), and GBP6 (log2 fc vs. VIM: 5.37; vs. DCM: 3.40) showed pronounced upregulation in ICIM vs. DCM and ICIM vs. VIM (Figure 2A,B).

### 3.3. Transcriptional Analysis Reveals a Potential Role of Interferon-Gamma in the Pathomechanism of ICIM

We compared the most differentially regulated genes between ICIM and DCM/VIM as well as between ICIM and patients receiving ICIs and negative results regarding the occurrence of myocarditis (Figure 2B). Here, the ICIM-specific gene program was absent. We were not able to observe a similar expression pattern in the ICI-group in genes that were upregulated neither in DCM nor in VIM. The clinical characteristics of the ICI-treated patients with rejected myocarditis diagnosis are shown in Appendix A.

ICIM-specific genes are involved in pathways that are mainly associated with biological processes of ‘cell division’, ‘chromosome segregation’, and ‘RNA splicing’. There is another pathway, namely ‘response to interferon-gamma’, that has no overlapping genes with any other pathways. This pathway includes GBP5 and GBP6 (Figure 2C). These two proteins belong to the most differentially upregulated ones between ICIM and the compared groups (Figure 2B).

Immunostainings for GBP5 and GBP6 revealed a high accumulation of the two proteins in cardiomyocytes of ICIM patients with a striped pattern. In the DCM patients’ biopsies, we found a low degree of GBP6 and GBP5 expression without a striped pattern. In VIM patients, a moderate degree of GBP5 and a low degree of GBP6 expression was found in the cardiomyocytes’ area. Both GBP5 and GBP6 co-localized with cardiomyocytes stained with α-Actinin, mainly in ICIM biopsies (Figure 2D).

### 3.4. CD8-Dependent Expression Patterns May Reveal ICIM Subtypes

According to the hypothesis that infiltration by CD8+ immune cells is the characteristic event in the pathophysiology of ICIM, we decided to group ICIM patients according to their median CD8 expression levels as well. Thereby, we found a set of 843 differentially regulated genes (FDR < 0.05) in ICIM patients (Appendix A). These genes partly overlapped with the genes that were uniquely upregulated in ICIM (73/843). There were 678 genes that were upregulated in ICIM patients with high CD8 expression, compared with 165 genes in patients with low CD8 expression (Appendix A). Pathway enrichment analysis of the upregulated CD8-dependent cluster showed three major pathways (neutrophil mediated immunity, *p* = 2.32 × 10^−34^; response to interferon-gamma, *p* = 1.45× 10^−26^; and leukocyte migration, *p* = 5.20× 10^−28^) that included the genes that we primarily identified as most regulated between ICIM and DCM or VIM (CXCL9, log2 fc: 5.89; CXCL11, log2 fc: 5.25; GBP5, log2 fc: 4.60) (Appendix A). In contrast, genes that were downregulated together with a low CD8 expression (*n* = 165) were mainly related to ATP-production and mitochondrial respiration (Appendix A).

Seeing high CD8 expression in the sequencing of the whole myocardial biopsies, we performed a bioinformatical approach to determine the degree of immune cell infiltration. We did not see large differences in the number of CD8+ cells between ICIM and control samples. The relative number of immune cells was relatively low compared with the predominance of fibroblasts and endothelial cells (Appendix A).

The identified expression subtypes of ICIM did not correlate significantly with either a distinct clinical course, biomarker elevation (hs-cTnT or NT-proBNP), or impairment of left ventricular function (Appendix A) in our cohort. However, we identified a tendency to high transcriptional levels of CD8, GBP5, GBP6, CXCL9, and CXCL11 with positive results of the CD3/CD8 immunostaining (Appendix A).

## 4. Discussion

### 4.1. Limitation by Clinical Variability of ICIM

The reported clinical cases show diverse cardiac manifestations of irAEs induced by immune checkpoint inhibitors. Even though these adverse effects are reported to be rare, they can be fatal [8,30]. This is why special attention must be paid to the diagnosis and therapy of ICIM. Given that the use of ICIs is rapidly increasing, we expect to face a substantial number of affected patients in the future [1].

Since there is no specific pathological test for ICIM yet, the diagnosis is still challenging and requires extensive cardiac imaging, cardiac biopsies, and subsequent interdisciplinary medical care.

In our cohort, clinical symptoms varied from pronounced HF and ACS-typical symptoms to completely asymptomatic presentations. Cardiac biomarkers such as hs-cTnT and NT-proBNP can be merely slightly elevated. On the other hand, we observed biomarker elevations, even in asymptomatic patients. These observations were in accordance with published data from case series and data from registries [8,10]. The detection of CD3/CD8 cells in cardiac biopsies is accepted as the gold standard for the diagnosis of ICIM [11]. Since cardiac biopsies are susceptible to sampling errors, there is still risk of underdiagnosis [26,31].

### 4.2. Strength of Transcriptional Profiling

We performed transcriptomic analysis to comprehensively describe gene expressions that potentially allow identification of unique factors in ICIM. To contrast, we chose cardiac diseases of comparable etiology (immune response in VIM) or symptoms (dyspnea, reduced ejection fraction in DCM). Additionally, we compared the identified pattern with patients who received ICIs, but the cardiological assessment did not reveal any evidence of ICIM.

Clustering of DCM, VIM, and ICIM revealed similarities between DCM and VIM in contrast to their correlation to ICIM. The stronger correlation in the PCA analysis and correlation heatmap between DCM and VIM may be explained by the initial severe reduction of the LVEF in the VIM patients.

Inhibition of the checkpoint pathway led to a distinct immune response compared with VIM, which was initially described in preclinical models of genetic PD-1 deletion [15]. The dominance of the infiltration by CD8+ cells vs. antibody-mediated effects on cardiac epitopes depended on genetic background in preclinical studies [16,17,18] and was not completely understood mechanistically.

In line with the hypothesis of the pathomechanisms of ICIM that are derived from preclinical models, we observed CD8 to be highly elevated in ICIM biopsies. We suppose that CD8 expression levels depended on the infiltration of immune cells. Nevertheless, the overall number of infiltrating cells, estimated via a bioinformatical approach, did not differ substantially between the different samples in ICIM, DCM, and VIM.

Among them, we found ICIM specimens with low and high CD8 levels. Those samples differed significantly in a whole set of genes. The difference of the CD8-dependent gene program can be explained by either different genetic predisposition or by a potential ‘second hit’ that drives antibody-mediated immune response in patients with low CD8. Low CD8 expression was associated with dysregulation of mitochondrial respiration, pointing towards a regulation of metabolic pathways.

Interestingly, the patients revealing high or low CD8 levels did not differ in CD3/CD8 positive or negative results with regard to histological sections. This observation may either be due to a sampling error, or it may be associated with a completely different molecular change that cannot be assessed by immunostaining. It remains unclear whether these different expression patterns are the cause or the consequence of the infiltration by CD8+ cells. Therefore, it will be necessary to establish more elaborated preclinical models or single cell-based analyses to answer the question of cellular interactions.

Apart from the CD8-dependent gene regulation, we observed a cluster of genes involved in the ‘response to interferon-gamma’ that is upregulated in ICIM.

Inflammatory genes that showed the highest fold changes between both DCM and VIM to ICIM are grouped in this interferon (IFN) gamma pathway (GBP5, GBP6).

IFN-gamma release is known to be a compensatory effect to mitigate the immune response of cytotoxic T cells by upregulation of PD-L1 in mice. Consequently, deletion of IFN-gamma led to a worsening of myocarditis [32]. The altered gene program that we observed in ICIM patients might be an adaption effect of excessive immune response in the myocardium. The involved TF programs displayed IFN-gamma-induced factors, e.g., IRF2 [33,34] and ETS [35,36]. The transcriptional response of cardiomyocytes to IFN-gamma may be a CD8-related or an additional mechanism by which ICIs mediate cardiac adverse events.

### 4.3. Potential Role of the Cardiac Inflammasome

The upregulation of IFN-gamma pathways, and particularly of GBP5, may indicate a possible link to the inflammasome pathway [37]. This class of GBP proteins is responsible for NLRP3 inflammasome assembly, which might be involved in the inflammatory response in ICIM patients. The link between PD-L1 and CD8-dependent activation of the inflammasome pathway has already been discussed to be the response of cancer cells upon ICI-treatment [38]. In immunostainings, we found an upregulation of GBP5 and GBP6 in cardiomyocytes of ICIM patients. Thus, we suggest a potential cardiomyocyte-specific regulation of the inflammasome in ICIM, which plays a minor role in DCM or VIM. Based on the transcriptional analysis, this process seems to be at least partly CD8-dependent and links leukocyte infiltration to an immune-response of the cardiomyocytes.

We were able to identify upstream regulators of the genes involved. These regulators are transcription factors that are known from pathological cardiac hypertrophy, e.g., YY1 [39], and fibrosis, e.g., SMAD binding motifs [40,41]. We propose a cardiac-specific regulation of a pathological gene program, which is mediated by the inflammasome and IFN-gamma in response to the inhibition of PD-1 or PD-L1 in ICIM patients.

## 5. Conclusions

ICIs have revolutionized the treatment of cancer. Immune-related adverse events such as ICIM appear to become a greater challenge due to their high mortality rates, especially when treating cancer in earlier stages. To date, there is a lack of diagnostic and therapeutic tools.

Our study reports transcriptional profiling of ICIM and reveals the involvement of CD8 and the IFN-gamma pathway. CD8 and CD8-dependent genes showed characteristic changes compared with DCM and VIM. IFN-gamma-dependent genes and activation of the cardiac inflammasome are unique parts of this gene program and may be additional targets for specific treatments. Pre-existing changes in the activity of the endogenous checkpoint pathways or inflammasome pathway might help to identify patients at risk.

## 6. Study Limitations

Since ICIM is a rare phenomenon, study populations are restricted to relatively small numbers and to retrospective analyses, case series, or registries. The inconsistent clinical presentation of our cohort and the reported cases can lead to underdiagnosis. Moreover, myocardial biopsies, the current gold standard for the diagnosis of ICIM, are susceptible to sampling errors.

Transcriptional profiling using RNA sequencing does not represent the actual protein expression in the heart, which can differ from transcript expression. We performed sequencing of whole myocardial biopsies, leaving the specific alterations of different cell types aside. 

In particular, we cannot completely rule out that parts of the identified CD8-dependent pattern are mainly derived from infiltrating lymphocytes. The degree of lymphocytic infiltration may vary either according to the disease activity or to the sampling location in the ventricle.

In addition, the sequenced biopsy and the biopsy used for histological assessment may not have been taken from the same location within the ventricle. This partially explains differences between the results of immunostaining and the expression pattern, even though all analysis were performed from samples of the same individual and at the same timepoint.

From a technical point of view, we cannot rule out batch effects caused by different subgroups, even though all samples were treated according to the same protocol in the same sequencing unit and all quality controls were passed.

## Figures and Tables

**Figure 1 cancers-13-02498-f001:**
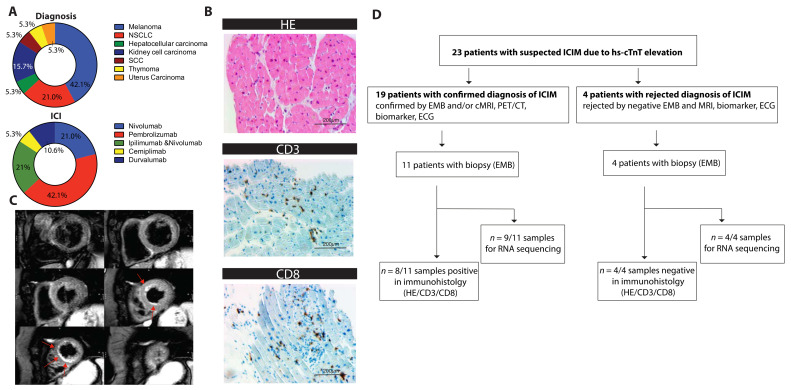
(**A**) Pie chart of the oncological diagnosis and the administered immune checkpoint inhibitors (ICIs) as indicated (*n* = 19). (**B**) Exemplary histological sections of pathological assessment showing lymphocyte infiltration. Staining for hematoxylin and eosin (HE) and immunostaining for CD3 and CD8 receptors is shown. (**C**) ICIM-compatible inflammation as apical accentuated edema in cMRI (T2 black blood images). Short axis of the left ventricle, six sections, shown from basal to apical, hyperintense edema is marked with red arrows. (**D**) Flowchart showing the numbers of patients who were screened due to hs-cTnT elevations, who were diagnosed with ICIM, who underwent biopsy, and who were assessed by immunohistology and/or by RNA sequencing. Biopsies were performed once the suspicion of ICIM was raised. CD3/8: cluster of differentiation 3/8, HE: hematoxylin and eosin; ICI: immune checkpoint inhibitor; NSCLC: non-small-cell lung cancer; SCC: squamous cell carcinoma.

**Figure 2 cancers-13-02498-f002:**
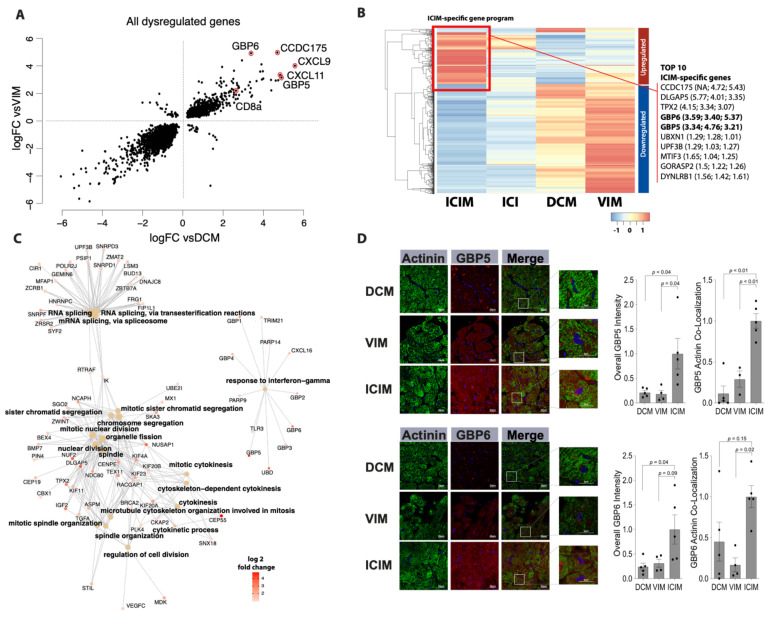
Differential gene expression in immune checkpoint inhibitor-associated myocarditis (ICIM, *n* = 9) in comparison with dilated cardiomyopathy (DCM, *n* = 11) and virus-induced myocarditis (VIM, *n* = 5) in cardiac biopsies (FDR < 0.05). (**A**) Concurrence of differential gene expression in ICIM related to DCM (vsDCM) and to VIM (vsVIM). The fold change (log2) of both comparisons is shown as a scatter plot. Transcripts of particularly elevated genes in both relations are highlighted as indicated. (**B**) Heatmap of differentially regulated genes in ICIM in comparison with DCM and VIM with a log2 fold change >1 or <−1. The gene expression is shown as *z*-values for ICIM (*n* = 9), ICI-treated patients without proof of ICIM (ICI, *n* = 4), DCM (*n* = 11), and VIM (*n* = 5). The cluster of genes that is upregulated the most in ICIM is marked with a red square as ‘ICIM-specific gene program’. The top 10 genes of this gene program are listed on the right side. In brackets, the log2 fold change of ICIM to ICI, DCM, and VIM is shown. (**C**) Network of enriched genes in ICIM with a log2 fold change > 1. The major Gene Ontology (GO) pathways and their corresponding genes are shown. Fold change (log2) of the genes (ICIM vs. VIM) is shown in red as indicated. (**D**) Guanylate binding protein 5 (GBP5) and guanylate binding protein 6 (GBP6) immunostaining of myocardial biopsies in ICIM (*n* = 5), DCM (*n* = 5), and VIM (*n* = 4) patients. Exemplary confocal images of GBP5- and GBP6-immunostaining (red) are shown for the three groups as indicated. Cardiomyocytes were stained with anti-Actinin (green). DAPI was used for staining of the nuclei. The total intensity of GBP5 and GBP6 and the corresponding overlap with Actinin were quantified. The intensity was normalized to the ICIM group. Columns show mean values with standard error of the mean (SEM) and individual values. *p*-values as indicated. ANOVA with Bonferroni correction for multiple testing was applied. DCM: dilated cardiomyopathy; ICI: immune checkpoint inhibitor; ICIM: ICI-associated myocarditis; VIM: virus-induced myocarditis.

**Table 1 cancers-13-02498-t001:** Table of ICIM patients’ characteristics, subgrouped according to the patients included in the RNAseq analysis. Hs-cTnT, NT-proBNP, and CK are specified as median values with interquartile range; the initial and maximum values are shown. BMI: body mass index, CK: creatine kinase, cMRI: cardiac magnetic resonance imaging, CRP: C-reactive protein, Hb: hemoglobin, HCC: hepatocellular carcinoma, Hs-cTnT: high sensitivity cardiac troponin T, NSCLC: non-small-cell lung cancer, NT-proBNP: N-terminal B-natriuretic propeptide, SCC: squamous-cell carcinoma, LV: left ventricle, LVEF: left ventricular ejection fraction, RV: right ventricle.

Characteristic	Total	RNA-Seq
*n*	19	9
Age (median, IQR)	75 (60.5, 78.5)	73.5 (61.25, 78.25)
Gender (male)	11 (57.8%)	5 (55.6%)
BMI (median, IQR)	25.2 (24, 28.3)	26.6 (25, 29.3)
Arterial hypertension	11 (57.9%)	6 (66.7%)
Diabetes	2 (10.5%)	2 (22.2%)
Hb (median, IQR)	13.1 (11.1, 13.5)	12.7 (11, 13.5)
Creatinine (median, IQR)	0.9 (0.73, 1.1)	0.78 (0.49, 1.1)
CRP (median, IQR)	13.8 (3.4, 34.8)	13.8 (3.8, 30.3)
Melanoma	8 (42.1%)	3 (33.3%)
NSCLC	4 (21.1%)	3 (33.3%)
HCC	1 (5.3%)	1 (11.1%)
Kidney cell carcinoma	3 (15.8%)	1 (11.1%)
SCC	1 (5.3%)	0 (0%)
Thymoma	1 (5.3%)	1 (11.1%)
Uterus carcinoma	1 (5.3%)	0 (0%)
Nivolumab	4 (21.1%)	4 (44.4%)
Nivolumab/Ipilimumab	4 (21.1%)	0 (0%)
Pembrolizumab	8 (42.1%)	3 (33.3%)
Cemiplimab	1 (5.3%)	0 (0%)
Durvalumab	2 (10.5%)	2 (22.2%)
LVEF > 50%	9 (47.4%)	5 (55.6%)
LVEF 40–50%	4 (21.1%)	1 (11.1%)
LVEF < 40%	6 (31.6%)	3 (33.3%)
Abnormal ECG	16 (84.2%)	8 (88.9%)
Initial Hs-cTnT (ng/l) (median, IQR)	55 (24.5, 618)	226 (24, 613)
Initial NT-proBNP (ng/l) (median, IQR)	1034 (723, 2900)	955 (397, 1942)
Initial CK (U/l) (median, IQR)	323 (97.5, 924.5)	630 (169, 2008)
Max Hs-cTnT (ng/l) (median, IQR)	278 (58, 1837.5)	681 (57, 1771)
Max NT-proBNP (ng/l) (median, IQR)	3309 (1145–7214)	2010 (1145, 5627)
Max CK (U/l) (median, IQR)	628 (192–1543)	630 (169, 2008)
Positive biopsy result	8 (42.1%)	6 (66.7%)
LV-biopsies	9 (47.4%)	8 (88.9%)
RV-biopsies	2 (10.5%)	1 (11.1%)
Positive cMRI/PET-CT	12 (63.2%)	6 (66.7%)
Definite ICIM diagnosis (1)	16 (84.2%)	7 (77.8%)
Probable ICIM diagnosis (1)	3 (15.8%)	2 (22.2%)

(1) According to the criteria of ICIM, published by Bonaca et al.

**Table 2 cancers-13-02498-t002:** Table of the DCM and VIM patients’ characteristics. Hs-cTnT and NT-proBNP are specified as median values with interquartile range; the initial and maximum values are shown. BMI: body mass index, cMRI: cardiac magnetic resonance imaging, CHD: coronary heart disease, CRP: C-reactive protein, DCM: dilated cardiomyopathy, Hb: hemoglobin, Hs-cTnT: high sensitivity cardiac troponin T, LGE: late gadolinium enhancement, LVEF: left ventricular ejection fraction, NT-proBNP: N-terminal B-natriuretic propeptide, VIM: virus-induced myocarditis.

Characteristic	VIM	DCM
*n*	5	11
Age (median, IQR)	22 (22, 42)	60 (52.5, 62)
Gender (male)	5 (100%)	7 (63.6%)
BMI (median, IQR)	23.7 (22.5; 25)	30.4 (21.9; 34.1)
Arterial hypertension	0%	81.8%
Diabetes	0%	100%
Hb (median, IQR)	15.5 (14.9, 15.8)	13.1 (12.65, 13.65)
Creatinine (median, IQR)	0.71(0.65, 0.73)	1.45 (1.05, 1.87)
CRP (median, IQR)	63.8 (37.1, 81)	3.7 (2, 11.1)
Hs-cTnT (ng/l) (median, IQR)	2545 (1456, 2813)	66.5 (24.5, 253.5)
NT-proBNP (ng/l) (median, IQR)	2201 (1, 2218)	1293 (468, 9414)
LVEF (initial) (1)		
preserved (>50%)	0%	0%
reduced (<30%)	100%	100%
LVEF (recent)		
preserved (>50%)	100%	0%
reduced (<30%)	0%	100%
Positive cMRI (edema, LGE)	100%	NA
Angiography (Positive for CHD)	0%	0%
Histology (positive for VIM/DCM)	100%	100%

(1): Timepoint of biopsy.

## Data Availability

RNAseq data of ICI patients are available in the EBI ArrayExpress Database, accession number E-MTAB-8867. The raw data files of DCM and VIM patients are available in the Gene Expression Omnibus Database (GEO-NCBI, accession number: GSE120567) as previously published [29].

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
