# Peer review of "Comparative Transcriptomics of Immune Checkpoint Inhibitor Myocarditis Identifies Guanylate Binding Protein 5 and 6 Dysregulation"

_cancers, 2021, doi:10.3390/cancers13102498_

Round 1

Reviewer 1 Report

Dear Authors,

I read your work titled "Comparative transcriptomucs of immune checkpoint inhibitor myocarditis identifies guanylate binding protein 5 and 6 dysregulation" with interest. I think this is a relevant body of work that would be of interest to oncologists and cardiologists alike. I have a few questions/clarifications:

1). You state that some patients were asymptomatic. How did they come to clinical attention then? Are these patients being monitored routinely with serial ECGs and echoes and that is how they were identified? If yes, please clarify in your methods.

2). There were 19 patients with 11 undergoing biopsies which verified the diagnosis in 8 patients but RNA extraction and sequencing was done in 9? Please clarify this discrepancy and the basis for choosing which patients to include in the analysis.

3). Please add a flowchart to your figures to show the number of patients screened, diagnosed and then the number sequencing was performed in to make the study flow easier to understand.

4). For the many reasons you yourself listed in study limitations, specifically "not being able to rule out that the transcriptome could have been derived from infiltrating lymphocytes" necessitates that you temper your conclusion in line 48-50.

5). Why was alpha-actinin staining used to localize GBP5 and 6 to cardiomyocytes? Why not use a more specific marker like anti-troponin antibody to definitely show enhanced expression in cardiomyocytes. If histological sections are available, this additional piece of data would strengthen the paper.

Reviewer 2 Report

This study revealed differential gene expression in biopsy samples of ICI-associated myocarditis. The identification of upregulation of IFNg and GBP5 and 6 genes are biologically intriguing, however, this manuscript will be improved f the authors could address following points.

  • Potential prediction value. Having less invasive assessment such as blood draw would facilitate the prediction of the onset of the ICI-associated myocarditis. If the authors could check blood samples and measure the level of inflammatory cytokines related with IFNg?
  • Minor issue. Could the author explain how to calculate the error bars in a panel D of figure 2? Errors bars of GBP6 intensity of ICIM group is seemingly too small, instead of a large distribution of actual data points.

Round 2

Reviewer 1 Report

Dear authors,

Thank you for providing/adding Figure 1 to show flow of patients. That clarifies the methodology significantly. After a spell/grammar check, I believe this paper is acceptable for publication.